# The Suffering of Advanced Chronic Renal Patients and Their Relationship with Symptoms in Loja, Ecuador

**DOI:** 10.3390/ijerph18105284

**Published:** 2021-05-16

**Authors:** Patricia Bonilla-Sierra, Ana Magdalena Vargas-Martínez, Fatima Leon-Larios, Joselin Valeria Arciniega Carrión, Tatiana Cecibel Jiménez Alverca, María de las Mercedes Lomas-Campos, José Rafael González-López

**Affiliations:** 1Health Sciences Department, Private Technical University of Loja, Loja 110107, Ecuador; pbonilla65@utpl.edu.ec (P.B.-S.); y8jvac@outlook.com (J.V.A.C.); tatysjimenez@outlook.com (T.C.J.A.); 2Nursing Department, Faculty of Nursing, Physiotherapy and Podiatry, Universidad de Sevilla, C/Avenzoar nº6, 41009 Seville, Spain; mlomas@us.es (M.d.l.M.L.-C.); joserafael@us.es (J.R.G.-L.)

**Keywords:** palliative care, chronic renal disease, symptoms

## Abstract

Advanced Chronic Kidney Disease (ACKD) supposes a public health problem in Ecuador that requires a comprehensive approach. In view of the scarcity of studies on the subject in this country, the objective of this research was to determine the signs and symptoms associated with the patients’ physical, social and psychological spheres that allow properly developing palliative care. A longitudinal, prospective and observational study was conducted with ACKD patients. In order to assess the symptomatic burden and suffering of these patients, the Edmonton Symptom Assessment System Revised: renal (ESAS-r) for renal patients and the Distress Thermometer (DT) were used. The sample consisted of a total of 246 patients. The most common symptoms that affect them, causing them suffering in their daily lives, are those related to well-being, difficulty falling asleep and itching. It is necessary that health professionals adapt care measures and help patients undergoing renal treatment, especially those who have suffered the disease for a longer period of time, in order to alleviate the patients’ suffering and therefore improve their daily lives. To such an end, a care plan could be designed that includes early palliative care.

## 1. Introduction

According to the National Renal Health Plan, in Ecuador, more than 65% of diabetes and arterial hypertension cases evolve to kidney failure. The prevalence rate of chronic kidney disease in advanced stages is 215.7 per million inhabitants. In Ecuador, there are approximately 17,500 patients with ACKD undergoing renal replacement treatment [1]. 

ACKD has a high prevalence to the extent of having become a public health problem in the last few years, especially in low- and middle-income countries and, despite this, no interdisciplinary approach is conducted with patients in most dialysis units. It is known that this treatment is not curative, which is why patients undergo a constant adaptation process to the frequent changes in their health status; therefore, the need to establish palliative care can be a reality. However, there are no well-established palliative care pathways [2].

This disease does not only morphologically and functionally affect the kidneys, but also the patients’ biopsychosocial spheres, subjecting them to physical, psychological and socio-labor limitations [3] and increasing their end-of-life medicalization [4]. In addition to the aforementioned, they experience very stressful situations such as diagnosis acceptance, disease progression, choice of treatment options and coping with dialysis [5,6,7,8]. 

The symptom burden in advanced kidney disease is similar to that of advanced cancer; identifying the physical, emotional, social and spiritual symptoms must be a priority. However, recognizing symptoms and being able to access early palliative care has not been an easy task, as patients do not possess adequate symptom management and often die in severe suffering related to the disease [9]. An alternative is comprehensive care centered on the patients’ and their needs, and not on the life prognosis. The choice should be joint work between the specialist in kidney disease and the palliative care team, that is to say, early and comprehensive palliative care [2,10,11]. 

We have engaged in this study in view of the limited knowledge about symptoms and suffering in patients with ACKD undergoing hemodialysis treatment in Ecuador, and particularly in the city of Loja, which is reflected in the scarcity of bibliography and research studies. The purpose was to know the most frequent symptoms, not only at the physical but also at the emotional level in the form of suffering, which might be exerting some influence on the quality of life of these patients. In addition, it was also of interest to analyze the associated clinical and sociodemographic factors. This will allow us to develop a palliative care plan adapted to the real needs of the patients. 

## 2. Materials and Methods

### 2.1. Study Design and Settings

A longitudinal, prospective and observational study was carried out. The study was conducted in the city of Loja, Ecuador. The participants were recruited in the centers of the Nefroloja Renal Unit and the Nephrology Centers of the Isidro Ayora Hospital and the Cornelio Samaniego Clinic in Loja. 

### 2.2. Sampling and Inclusion

Consecutive and systematic sampling was conducted with patients who attended the hemodialysis centers during the period from January to June 2019. The following were established as inclusion criteria: patients over 18 years old with a minimum period on hemodialysis treatment (at least 3 months), with their cognitive abilities preserved and who agreed to freely participate in the study. Exclusion criteria defined patients who refused to participate in the study or presented some pre-existing psychological or psychiatric disorder. 

### 2.3. Measures

The following sociodemographic variables were collected: gender, age, marital status, religion, occupation, schooling level, personal background, clinical time of the disease and time undergoing treatment. In order to assess the symptomatic burden and suffering of these patients, the ESAS-r for renal patients and the DT were used. These scales were applied at the baseline moment and after 2 months. Concomitant diseases were recruited from medical records. Most often, diseases were arterial hypertension and any other cardiac disease, diabetes mellitus, hypothyroidism, chronic obstructive pulmonary disease and cancer. 

The ESAS-r is a questionnaire for the quantitative assessment of symptoms that allows documenting multiple symptoms reported by the patient, such as: pain, depression, anxiety, nausea, affected well-being, changed appetite, sleepiness, shortness of breath and fatigue, in addition to other symptoms for patients with chronic kidney disease, such as sleep and pruritus [12,13]. By means of a numerical and visual scale, the patient identifies each of the symptoms from 0 to 10. Subsequently, a clinical profile of the symptoms and the disease progression are determined, which allows establishing the type of care required by the person. In 2011, Watanabe et al. [14] introduced a reviewed version including more terms, such as exhaustion, discouragement and anxiety. They established an orderly relationship of the symptoms, a new format and written instructions for the time reference of the period during which symptoms are assessed [15]. This instrument has been validated to assess the symptoms of renal patients and their health-associated quality of life [12]. Moderate intensity of any symptom is defined between 4 and 6, and severe intensity between 7 and 10 on a Likert scale. The total score of the symptoms is calculated by adding up the scores of the 10 symptoms, obtaining a total between 0 and 100 points. 

The DT is a tool validated in people with chronic kidney disease [16] that includes a numerical and visual scale assessing the level of malaise or distress by means of a single item on an 11-point scale from 0 to 10, where 0 is the absence of distress and 10 is extreme distress, as well as a list of the problems associated with the suffering classified in 4 domains: practical problems (*n* = 5, child care, insurance/financial, transportation, work/school), family problems (*n* = 2, dealing with one’s partner, dealing with children), emotional problems (*n* = 7, depression, fear, nervousness, sadness, worry, loss of interest in usual activities, spiritual/religious concerns) and physical problems (*n* = 21, appearance, bathing/dressing, breathing, changes in urination, constipation, diarrhea, eating, fatigue, feeling swollen, fevers, getting around, indigestion, memory/concentration, mouth sores, nausea, dry/congested nose, pain, sexuality, dry/itchy skin, sleep, tingling in hands/feet) [17]. The patients indicated if they had experienced any of these 35 problems in the previous week.

### 2.4. Statistical Analysis

The descriptive characteristics of the subjects related to sociodemographic aspects, treatment (hemodialysis) and comorbidities were reported as mean values, standard deviation -SD-, or n (%) of the total. Additionally, a comparison of means and frequencies by the hospital, clinic or service attended by the patient (Cornelio Samaniego Clinic, Isidro Ayora Hospital, Nefroloja Hemodialysis Service) was reported by *p*-values after using a t-test, ANOVA or Pearson’s χ^2^, depending on the type of variable. An analysis was conducted of the differences in the symptoms scores at baseline and at 2-month follow-up, measured through the Edmonton Symptoms Assessment System Revised and the DT, as well as the differences in symptoms scores by the number of concomitant disorders at the baseline period. For this, parametric tests (t-test or ANOVA) and non-parametric tests (Kruskal–Wallis and Wilcoxon) were used when the variable was numerical and depending on its normality. The factors associated with the number of problems perceived through the DT were studied using linear regression analysis. The backward strategy was used to obtain the most parsimonious model.

Some variables such as age were centered using medians because a value of 0 was not included in these variables for the sample under study. Data was entered into R software (R-3.6.3 version, Free Software Foundation’s GNU General Public License: https://www.r-project.org/about.html, accessed on 30 November 2020) for statistical analysis. A *p*-value < 0.05 was accepted as statistically significant.

### 2.5. Ethical Considerations

All participants were informed about the study objectives and were asked to sign a written informed consent of their participation. They were guaranteed anonymity and data confidentiality in the information provided. This study was approved by the Research Committee of the Private University of Loja, Ecuador, under reference number 2019015G. 

## 3. Results

### 3.1. Descriptive Characteristics of the Sample

Initially, a total of 258 subjects comprised the sample. Of these, 12 individuals, who came from the Nefroloja Hemodialysis Service, did not provide data in relation to the follow-up questionnaire in which the Edmonton Symptoms Assessment System Revised and the DT were administered, resulting in a total of 246 subjects for a response rate of 95.3%.

In relation to sociodemographic characteristics, the mean age was 61.05 (SD 13.76) years old with a range between 19 and 91 years old, and 75% of the subjects were over 54 years old. Statistically significant differences related to age were found among the patients that attended the Cornelio Samaniego Clinic, the Isidro Ayora Hospital and the Nefroloja Hemodialysis Service, with the highest mean age being found in the latter group of patients. Approximately half of the subjects were men and the other half were women, finding no statistically significant difference by hospital, service or clinic attended by the subjects. The proportion of women was slightly lower than that of men in the total sample and among those who attended the Isidro Ayora Hospital, with the exception of those who attended the Cornelio Samaniego Clinic, where 65.1% were men, and those who attended the Nefroloja Hemodialysis Service, where half were men and half women. Most of the subjects identified themselves with the Catholic religion, accounting for 91% in all cases. Statistically significant differences were found in relation to occupational status. In general, the highest percentage of subjects were unemployed, exceeding 50% in all cases except in the Cornelio Samaniego Clinic, where 46% of the individuals were retired and the percentage of unemployed people did not reach 30%. In addition, in this clinic, a higher percentage of people in active employment was observed, equaling the proportion of unemployed individuals. More than half of our sample (57.5%) had basic/primary studies (Appendix A, Table A1). 

Concerning the characteristics related to treatment (hemodialysis), the mean diagnosis time was 5.41 (SD 5.47) years, with 75% of the subjects having a diagnosis time of fewer than 7 years. A statistically significant difference was found by hospital, clinic or service, with the highest mean diagnosis time being found among the subjects who attended the Cornelio Samaniego Clinic. In relation to the mean hemodialysis treatment time, 75% of the subjects had not been in treatment for more than 5 years (Appendix A, Table A1).

Regarding the cause of chronic kidney disease, Type II Diabetes Mellitus was the main cause in most of the subjects, accounting for more than 50% in all cases. In addition to chronic kidney disease, other concomitant diseases were explored in the sample, finding statistically significant differences by the hospital, clinic or service attended by the study patients. The Nefroloja Hemodialysis Service was the place where a higher proportion of patients had 2 or more comorbidities, reaching 47.5% (Appendix A, Table A1).

### 3.2. Edmonton Symptoms Assessment System Revised and the DT at Baseline and 2-Month Follow-Up

When the symptoms scores measured with the Edmonton Symptoms Assessment System Revised at baseline and 2-month follow-up were compared, the only symptoms that presented statistically significant differences were well-being, difficulty falling asleep and itching, with a higher score in all cases in the follow-up period, which means worsening of these symptoms (Table 1).

In relation to the problems assessed through the DT in the previous week, the number of practical and family problems showed a statistically significant increase after 2 months of assessment. In general, although the “other problems” categories did not present any statistically significant difference, patients indicated an increase in the number of problems over the 2-month follow-up period (Table 1).

### 3.3. Edmonton Symptoms Assessment System Revised and the DT by Number of Concomitant Disorders at the Baseline Period

Concerning the differences found in symptoms scores according to the Edmonton Symptoms Assessment System Revised by the number of comorbidities at the baseline period, the only symptom that presented a statistically significant difference was nausea, which reached the highest score among those subjects who had two concomitant diseases. In general, the rest of the symptoms worsened as the number of diseases increased, although this did not happen in all cases, and no statistically significant difference was found (Table 2).

In relation to the number of problems in the past week assessed with the DT, statistically significant differences were found regarding emotional, physical and total problems, finding a greater number of problems as the number of comorbidities increased (Table 2).

### 3.4. Factors Associated with the Number of Problems Perceived According to the DT

Prior to using a linear regression model to analyze the factors associated with distress level, a correlation analysis between the numerical variables was carried out, presented in Table 3. Initially, when analyzing the factors associated with the total number of problems at 2-month follow-up through a linear regression model, it was found that the number of problems was reduced by 1.75 times in men when compared to women. However, those who had primary or secondary studies and those who attended the Isidro Ayora Hospital or the Cornelio Samaniego Clinic, as well as those with Type II Diabetes Mellitus as the cause of chronic kidney disease, had the number of problems reduced compared to those who had no studies, those who attended the Nefroloja Hemodialysis Service and those with arterial high blood pressure as the cause of chronic kidney disease. Conversely, the total number of problems increased by 0.41 times as the hemodialysis treatment time increased (Table 4).

When the backward strategy was applied and the correlation analysis among regressors was taken into account, the resulting model found the same statistically significant differences related to the predictors included. This last model explained 28.1% of the total number of problems at 2-month follow-up, finding no statistically significant difference in comparison to the model which included all of the predictors (Table 4).

Additional analyses were replicated in subgroups (Cornelio Samaniego, Isidro Ayora Hospital, Nefroloja). These findings are shown in Table 5. The only model that was statistically significant was the one performed in patients attending the Nefroloja Hemodialysis Service, given the sample size. For these patients, the total number of problems at 2-month follow-up was associated with gender.

## 4. Discussion

The purpose of this research was to determine the symptomatic burden and problems derived from chronic kidney disease during hemodialysis treatment and its evolution over time, as well as its relation with emotional, family and practical aspects that suppose emotional burden and distress, with the possibility of resulting in suffering. As reported by Damery et al. [18], detecting suffering in patients with chronic kidney disease, and especially in those in its final stages, is important to ensure optimum care. In addition, they suggest that reducing the number of emotional and psychological problems can improve the well-being and coping ability of these patients for complex treatments [19]. As indicated by the results found, the participants mentioned a wide symptomatic burden, especially in the aspects related to well-being, falling asleep and pruritus. This study developed an understanding of the symptomatology associated with patients with renal problems from a southern Ecuadorian area, being one of the first studies published in the country. 

Our results are in line with those offered by other authors [20,21], also observing the greater presence of symptoms when there is more associated morbidity [22,23], with even greater impairment depending on the disease [24] being multi-pathological and the complexity of patients. It was also observed that the evolution of the disease has an influence on symptomatic burden; consequently, patients with a longer time of disease diagnosis and undergoing hemodialysis treatment suffer more than those with a more recent diagnosis [6,25]. 

Gender exerts an influence on the greater impairment imposed by the disease, with worse scores in women as time passes by [19]. Although chronic kidney disease is more frequent in men, women present more associated symptoms. This has also been found in other previous studies [18,26,27].

Suffering in patients with chronic kidney disease is directly related to symptomatic burden [28]. Regarding the list of problems intervening in the suffering of study patients, and as already reflected in the results, a higher intensity was observed in physical problems [7], followed by emotional, practical and family problems to a lesser extent. In the study conducted by Franco et al. [29], it was also observed that family problems are the least frequent. 

Palliative care can be an alternative for patients when they present many symptoms [30]. As the disease progresses, palliative care that mitigates the patients’ suffering is needed, improving quality of life both in patients and families [31]. Whereas palliative care is indeed integrated in developed countries, there are still a few developing countries that have implemented it [32]. For this, it is necessary to perform a symptomatic control that allows for the early development of a palliative care plan [2,33]. Chronic kidney disease has been associated with a low quality of life [34,35]; however, it has been proven that palliative care can enhance it [36,37]. In Ecuador, the current situation of palliative care provided to renal patients has led to the proposal of its integration in the academic training of future health professionals, as suggested by Lam et al. [4].

In Ecuador, a series of changes have taken place to favor the integration of palliative care in the structure of the National Health System; however, there are major deficiencies related to the availability of medications (especially of immediate-release opioids), limited service provision and an insufficient number of professionals due to scarce undergraduate and graduate training [38], with service provision for patients with ACKD being almost nonexistent.

### Limitations

Among the limitations of this study are the sample selection by convenience and the fact that the study was conducted in a single city of the country. Nevertheless, the participants were recruited from up to three centers of the city, obtaining a broad sample, which could overcome some deficiencies of the non-probabilistic sampling. This does not allow generalizing the results to other contexts and the general population presenting the same characteristics, but sheds light on the suffering of these patients in Ecuador, specifically in the city of Loja. 

Another limitation is related to the absence of other variables related to the level of anguish/suffering in these patients that have been studied by other authors [18,39], such as the health professionals–patient ratio or the service model used.

## 5. Conclusions

The results provided by this research provide knowledge regarding the most common symptoms in renal patients that cause them suffering in their daily lives, such as well-being, difficulty falling asleep and itching. It is necessary that health professionals adapt care measures and help patients undergoing renal treatment, especially those who have suffered the disease for a longer period of time, in order to alleviate suffering and therefore improve the patients’ daily lives. To such an end, a care plan could be designed that includes early palliative care. This study will help to develop a palliative care plan that assists the most impaired functions in chronic renal patients. 

## Figures and Tables

**Table 1 ijerph-18-05284-t001:** Differences in symptoms scores (at baseline and 2-month follow-up) according to the Edmonton Symptoms Assessment System Revised and the DT (*n* = 246).

Scales/Tools	Baseline Mean (SD)	2-Month Follow-Up Mean (SD)	*p*-Value
Edmonton Symptoms			
Pain	4.18 (3.38)	4.42 (3.17)	0.108
Fatigue	3.71 (3.12)	3.83 (2.95)	0.447
Nausea	1.75 (2.65)	1.88 (2.59)	0.328
Depression	3.49 (3.48)	3.79 (3.28)	0.061
Anxiety	2.70 (3.13)	2.63 (3.00)	0.663
Sleepiness	2.76 (2.94)	2.92 (2.78)	0.256
Appetite	2.22 (2.59)	2.57 (2.36)	0.050
Well-being	2.18 (2.00)	3.03 (2.07)	0.000
Dyspnea	1.26 (2.32)	1.39 (2.27)	0.327
Difficulty Falling Asleep	3.27 (3.38)	3.78 (3.21)	0.001
Itching	3.13 (3.27)	3.44 (3.01)	0.032
DT			
Practical Problems	1.42 (1.31)	1.69 (1.25)	0.003
Family Problems	0.31 (0.61)	0.45 (0.69)	0.001
Emotional Problems	3.45 (2.37)	3.59 (1.96)	0.355
Physical Problems	8.39 (4.11)	8.45 (4.17)	0.824
Total DT Score	13.57 (6.69)	14.18 (6.71)	0.202

**Table 2 ijerph-18-05284-t002:** Differences in symptom scores according to the Edmonton Symptoms Assessment System Revised and the DT by the number of comorbidities at the baseline period (*n* = 246).

Scales/Tools	No Disorder Mean (SD)	1 Disorder Mean (SD)	2 Disorders Mean (SD)	3 or More Disorders Mean (SD)	I-Value
Edmonton Symptoms					
Pain	3.82 (3.46)	4.57 (3.35)	4.35 (3.39)	3.39 (3.05)	0.354
Fatigue	3.34 (3.16)	4.08 (3.31)	3.46 (3.01)	4.28 (2.02)	0.343
Nausea	1.16 (2.18)	1.54 (2.58)	2.80 (3.16)	2.33 (2.52)	0.003
Depression	3.13 (3.45)	3.43 (3.42)	4.17 (3.94)	3.39 (2.12)	0.402
Anxiety	2.37 (2.95)	2.65 (3.34)	3.17 (3.27)	3.00 (2.38)	0.512
Sleepiness	3.17 (3.11)	2.55 (2.98)	2.31 (2.70)	3.28 (2.56)	0.281
Appetite	2.06 (2.51)	2.36 (2.89)	2.17 (2.26)	2.33 (2.45)	0.891
Well-being	2.29 (2.11)	2.23 (2.07)	1.91 (1.57)	2.22 (2.39)	0.726
Dyspnea	0.74 (1.56)	1.42 (2.51)	1.57 (2.63)	1.83 (2.94)	0.085
Difficulty Falling Asleep	2.71 (3.43)	3.52 (3.43)	3.59 (3.19)	3.56 (3.48)	0.336
Itching	2.96 (3.27)	3.35 (3.32)	2.63 (3.11)	4.28 (3.30)	0.250
DT					
Practical Problems	1.35 (1.27)	1.40 (1.41)	1.41 (1.25)	1.89 (1.60)	0.468
Family Problems	0.29 (0.62)	0.32 (0.59)	0.31 (0.61)	0.39 (0.70)	0.947
Emotional Problems	3.10 (2.12)	3.39 (2.41)	3.63 (2.54)	4.78 (2.37)	0.048
Physical Problems	7.17 (4.11)	8.26 (3.94)	9.76 (3.99)	10.44 (3.60)	0.000
Total DT Score	11.91 (6.51)	13.37 (6.51)	15.11 (6.53)	17.50 (6.78)	0.002

**Table 3 ijerph-18-05284-t003:** Correlations among regressors.

	DT at 2-Month Follow-Up	Age	Number of Concomitant Disorders	Diagnosis Time	Hemodialysis Treatment Time	Estimated Time of Arrival at the Hospital/Clinic	Number of Weekly Sessions (Hemodialysis)
DT at 2-Month Follow-up	1.000						
Age	0.106	1.000					
Number of Concomitant Disorders	0.262 ***	0.228 ***	1.000				
Diagnosis Time	−0.010	0.033	−0.048	1.000			
Hemodialysis Treatment Time	0.144 **	−0.040	0.025	0.758 ***	1.000		
Estimated Time of Arrival at the Hospital/Clinic	0.052	0.025	0.008	−0.126 ***	−0.115 *	1.000	
Number of Weekly Sessions (Hemodialysis)	−0.093	−0.117 **	−0.161 **	0.030	−0.003	−0.068	1.000

Significance level: *** *p*-value < 0.001, ** *p*-value < 0.01, * *p*-value < 0.05.

**Table 4 ijerph-18-05284-t004:** Factors associated with the DT in patients undergoing hemodialysis treatment (at 2-month follow-up).

Total Number of Problems at 2-Month Follow-Up (DT)	Model 1 (All Predictors)	Model 2 (Backward Model)
	Coefficient (95% CI)	Coefficient (95% CI)
Age (Centered to 63 Years Old)	0.000 (−0.07, 0.07)	0.007 (−0.06, 0.07)
Gender [Male]	−1.754 (−3.40, −0.11) *	−1.790 (−3.38, −0.20) *
Marital status [Married]		
Separated/Divorced	−0.098 (−2.84, 2.65)	
Single	0.555 (−1.60, 2.71)	
Widowed	0.433 (−2.05, 2.91)	
Religion [Catholic]	0.281 (−2.72, 3.28)	
Occupational Status [Actively Employed] Not Working/Unemployed Retiree	2.006 (−0.32, 4.33) 2.000 (−0.52, 4.52)	2.066 (−0.15, 4.29) 1.881 (−0.54, 4.30)
Schooling Level [No Studies]		
Primary Studies	−6.957 (−13.01, −0.90) *	−6.899 (−12.85, −0.94) *
Secondary Studies/Technology or Technical	−7.197 (−13.50, −0.89) *	−7.211 (−13.37, −1.05) *
Third Level/Tertiary/University Studies	−5.021 (−11.41, 1.37)	−5.094 (−11.33, 1.14)
Diagnosis Time (Years)	−0.162 (−0.39, 0.07)	−0.158 (−0.38, 0.07)
Hemodialysis Treatment time (Years)	0.410 (0.13, 0.69) **	0.409 (0.13, 0.68) **
Number of Weekly Sessions (Hemodialysis)	−1.426 (−6.46, 3.61)	−1.366 (−6.25, 3.52)
Estimated Time of Arrival at the Hospital/Clinic (Minutes)	0.003 (−0.01, 0.01)	
Use of Health Transport [No]	5.151 (−3.23, 13.53)	
Hospital/Clinic/Service [Nefroloja]		
Isidro Ayora Hospital	−4.315 (−6.63, −2.01) ***	−4.546 (−6.45, −2.44) ***
Cornelio Samaniego Clinic	−5.168 (−7.29, −3.05) ***	−5.014 (−7.02, −3.01) ***
Cause of Chronic Kidney Disease [High Blood Pressure]		
Type II Diabetes Mellitus	−0.054 (−1.89, 1.78)	
Type II Diabetes Mellitus and High Blood Pressure	−1.213 (−5.33, 2.90)	
Other Causes	−2.264 (−5.96, 1.43)	
Number of Concomitant Disorders	0.246 (−0.74, 1.23)	0.382 (−0.51, 1.27)

Model 1: R2 = 0.293, R2 adjusted = 0.223, *p* < 0.001; Model 2: R2 = 0.281, R2 adjusted = 0.2405, *p* < 0.001; significance level: *** *p*-value < 0.001, ** *p*-value < 0.01, * *p*-value < 0.05. CI: Confidence Interval.

**Table 5 ijerph-18-05284-t005:** Factors associated with the DT in patients undergoing hemodialysis treatment (by subgroups).

Total Number of Problems at 2-Month Follow-Up (DT)	Nefroloja	Isidro Ayora Hospital	Cornelio Samaniego Clinic
	Coefficient (95% CI)	Coefficient (95% CI)	Coefficient (95% CI)
Age (Centered to 63 Years Old)	0.01 (−0.09, 0.11)	0.04 (−0.09, 0.17)	−0.05 (−0.18, 0.09)
Gender [Male]	−2.55 (−4.82, −0.29) *	−1.16 (−4.54, 2.22)	−2.42 (−5.93, 1.08)
Occupational Status [Actively Employed] Not Working/Unemployed Retiree	2.79 (−0.48, 6.06) 3.72 (−0.08, 7.52)	3.61 (−0.88, 8.11) 0.11 (−7.67, 7.89)	−0.92 (−5.85, 3.99) −0.39 (−4.84, 4.06)
Schooling Level [No Studies]			
Primary Studies	−4.99 (−13.86, 3.87)	−1.52 (−15.88, 12.83)	−8.95 (−21.92, 4.02)
Secondary Studies/Technology or Technical	−8.46 (−16.94, 0.02)	1.47 (−13.06, 15.99)	−9.85 (−23.27, 3.57)
Third Level/Tertiary/University Studies	−7.70 (−15.89, 0.48)	−1.02 (−15.21, 13.16)	−6.99 (−20.02, 6.04)
Diagnosis Time (Years)	−0.19 (−3.47, 3.08)	0.10 (−0.36, 0.57)	−0.19 (−0.62, 0.23)
Hemodialysis Treatment Time (Years)	0.36 (−2.92, 3.64)	0.19 (−0.48, 0.85)	0.54 (0.03, 1.06) *
Number of Weekly Sessions (Hemodialysis)	−1.44 (−6.25, 3.36)		
Number of Concomitant Disorders	0.45 (−0.67, 1.58)	2.07 (−0.24, 4.38)	−1.40 (−3.65, 0.86)

R2 (Nefroloja) = 0.196, R2 adjusted (Nefroloja) = 0.116, *p* = 0.009; R2 (Isidro Ayora) = 0.156, R2 adjusted (Isidro Ayora) = 0.016, *p* = 0.536; R2 (Cornelio Samaniego) = 0.183, R2 adjusted (Cornelio Samaniego) = 0.022, *p* = 3532; significance level: * *p*-value < 0.05.

## Data Availability

The data that support the findings of this study are available from the corresponding author (A.-M.V.-M.) upon reasonable request.

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
