# Peer review of "The Suffering of Advanced Chronic Renal Patients and Their Relationship with Symptoms in Loja, Ecuador"

_ijerph, 2021, doi:10.3390/ijerph18105284_

Round 1

Reviewer 1 Report

This is an observational study of sufficient size that concerns physical and emotional Symptoms of patients with advanced renal disease.

Statistics is suitable, results are convincing, limitations are adequately discussed   

Author Response

Response to Reviewer 1 Comments

We are grateful to the reviewers for the time and effort they put in reviewing our paper. We are also thankful for the insightful comments, which led us to improve our research. Please, find below a response to each comment. For readability purposes, our responses to the comments are marked in italics.

Point 1: This is an observational study of sufficient size that concerns physical and emotional Symptoms of patients with advanced renal disease.

Statistics is suitable, results are convincing, limitations are adequately discussed  

Response 1:  Thanks for this encouraging comment and your review.

Reviewer 2 Report

The manuscript “Suffering of Advanced Chronic Renal Patients and its Relationship to Symptoms in Loja, Ecuador” by Patricia Bonilla-Sierra et al. evaluates the symptomatic and emotional burden of chronic kidney disease (CKD) in 246 patients on hemodialysis at baseline and after two months of follow-up. In this longitudinal study, the authors found that well-being, difficulty falling asleep, and itching are the most common symptoms causing suffering in this group of renal patients. Latin America has one of the highest death rates from CKD worldwide; hence this study is precious for providing information on Ecuadorian patients that could help tackle kidney disease in this region. I agree with the authors that interdisciplinary care plans for CKD patients that include palliative care are urgently needed, and this study gives evidence of that. Here are some comments to improve the manuscript.

In the Materials and methods section, the authors first describe their inclusion criteria as patients over 18 years old with a minimum period of 3 months on hemodialysis treatment. Later, they mention that the scales were applied at the baseline moment, when the patient initiates the hemodialysis treatment and after two months (Line 80-81). This is confusing since the patients are not initiating hemodialysis at baseline. I suggest rephrasing the sentence.

Throughout the manuscript, the authors refer to concomitant disorders and Table 3 shows the differences in symptoms scores according to the Edmonton Symptoms Assessment System Revised and The Distress Thermometer by number of concomitant disorders at the baseline period. However, it is unclear which are these concomitant disorders or how the authors gathered this information. This information is important to understand the significance of the results found in the study, I suggest adding it to the materials and methods and discussion sections.

In the discussion section, the authors mention that family problems intervene in the suffering of the study's patients to a lesser extent, an observation similar to that of Franco et al. (Lines 252-255). Do the authors think this is a specific reflection of the population and country where the study was carried out? Additionally, could the authors describe the components of a palliative care plan based on their observations?

Please check that all terms are abbreviated only when they are first mentioned and that abbreviations are consistent throughout the text. (Examples: Advanced Chronic Kidney Disease (ACKD) and The Distress Thermometer (DT))

Author Response

Response to Reviewer 2 Comments

We are grateful to the reviewers for the time and effort they spent reviewing our paper. We are also thankful for the insightful comments, which led us to improve our research. Please, find below a response to each comment. For readability purposes, our responses to the comments are in italics. Finally, changes in the manuscript are highlighted in yellow. We hope that this new version meets reviewer´s expectations. The current version of the manuscript has been reviewed by an English native proofreader.

Point 1:  The manuscript “Suffering of Advanced Chronic Renal Patients and its Relationship to Symptoms in Loja, Ecuador” by Patricia Bonilla-Sierra et al. evaluates the symptomatic and emotional burden of chronic kidney disease (CKD) in 246 patients on hemodialysis at baseline and after two months of follow-up. In this longitudinal study, the authors found that well-being, difficulty falling asleep, and itching are the most common symptoms causing suffering in this group of renal patients. Latin America has one of the highest death rates from CKD worldwide; hence this study is precious for providing information on Ecuadorian patients that could help tackle kidney disease in this region. I agree with the authors that interdisciplinary care plans for CKD patients that include palliative care are urgently needed, and this study gives evidence of that. Here are some comments to improve the manuscript.

Response 1:  Thanks for this encouraging comment and your review.

Point 2:  In the Materials and methods section, the authors first describe their inclusion criteria as patients over 18 years old with a minimum period of 3 months on hemodialysis treatment. Later, they mention that the scales were applied at the baseline moment, when the patient initiates the hemodialysis treatment and after two months (Line 80-81). This is confusing since the patients are not initiating hemodialysis at baseline. I suggest rephrasing the sentence.

Response 2:  Thanks for your comment. Although it could be a little bit confusing, they are two different things. Firstly, to be included in the sample, it is required that the patient was in the hemodialysis treatment for a period (in concrete, at least 3 months).

Once the patient meets the inclusion criteria, baseline data were collected. Two months later, the data would be collected again. In order to avoid misunderstandings, this part of the sentence “when the patient initiates the hemodialysis treatment” was deleted. This has been rewritten (lines 7071 and 7980).

Point 3: Throughout the manuscript, the authors refer to concomitant disorders and Table 3 shows the differences in symptoms scores according to the Edmonton Symptoms Assessment System Revised and The Distress Thermometer by number of concomitant disorders at the baseline period. However, it is unclear which are these concomitant disorders or how the authors gathered this information. This information is important to understand the significance of the results found in the study, I suggest adding it to the materials and methods and discussion sections.

Response 3:  

Thank you for this observation that was amended. Please, see lines in methods section: “Concomitant diseases were recruited from the medical records. More often diseases were arterial hypertension and any other cardiac disease, diabetes mellitus, hypothyroidism, chronic obstructive pulmonary disease and cancer” (lines 8083). In discussion section: “Our results are in line with those offered by other authors [20,21], also observing greater presence of symptoms when there is more associated morbidity [22,23], with even greater impairment depending on the disease [24] being multi-pathological and complex patients” (lines 258261).

Point 4:  In the discussion section, the authors mention that family problems intervene in the suffering of the study's patients to a lesser extent, an observation similar to that of Franco et al. (Lines 252-255). Do the authors think this is a specific reflection of the population and country where the study was carried out? Additionally, could the authors describe the components of a palliative care plan based on their observations?

Response 4: Authors think that there was a misunderstanding, our study suggests that the suffering in patients with chronic disease affect in less proportion to family problems. We think that the culture and the country may be influencing in the coping of the disease. Ecuadorian usually live very close to their extent family that support the daily care.

Regarding the second request, we suggest to read the following paragraph that describes the weaknesses of the palliative care: “Whereas palliative care is indeed integrated in developed countries, there are still few developing countries that have implemented it [32]. For this, it is necessary to perform a symptomatic control that allows for the early development of a palliative care plan [2,33]. Chronic kidney disease has been associated to low quality of life [34,35]; however, it has been proved that palliative care can enhance it [36,37]. In Ecuador, the current situation of palliative care provided to renal patients leads to the proposal of its integration in the academic training of the future health professionals, as suggested by Lam et al. [4].

In Ecuador, a series of changes has taken place to favor the integration of palliative care in the structure of the National Health System; however, there are major deficiencies related to the availability of medications, especially of immediate-release opioids, limited-service provision, and insufficient number of professionals due to scarce undergraduate and graduate training [38], with service provision for patients with ACKD being almost nonexistent” (lines 280294).

Point 5:  Please check that all terms are abbreviated only when they are first mentioned and that abbreviations are consistent throughout the text. (Examples: Advanced Chronic Kidney Disease (ACKD) and The Distress Thermometer (DT))

Response 5: Thank you for your suggestion. This has been checked throughout the manuscript.

Reviewer 3 Report

Short Summary

The research illustrates how ACKD does not only affect kidneys, but also patients' biopsychosocial sphere, introducing physical, psychological and socio-labor limitations, and increasing their end-of-life medicalization. To increase the limited knowledge about symptoms and suffering in patients with ACKD undergoing hemodialysis treatment in Ecuador, the research focuses on the most frequent symptoms, not only at the physical but also at the emotional level in the form of suffering, which might be exerting some influence on patients' quality of life. Specifically, the research illustrates the results of a longitudinal study involving a sample of 258 patients (246 respondents), disaggregating by hospital/clinique/service, by a baseline and a two months follow-up scenario, and by a number of concomitant diseases. Finally, a regression model (OLS following a backward stepwise procedure) is presented, explaining the total number of problems at 2-months follow up in terms of socioeconomic and health factors. Results illustrate how suffering in patients with chronic kidney disease is directly related to the symptomatic burden. More in general, chronic kidney disease has been associated to low quality of life, however, it has been proved that palliative care can enhance it.

Broad Comments

The research seems to be of interest to the readership of the Journal, however empirical analysis should be improved. Due to its descriptive use, the use of OLS technique is not questionable, but the model specification should be improved. As an example, a correlation analysis among regressors may be added, then the model may be developed on a subset of the available regressors to avoid multicollinearity, at most controlling for (some) socioeconomic factors. Furthermore, the R^2 is low, therefore interaction effects and non linear relations could be introduced to improve the goodness-of-fit. Finally, as there are multiple regressors, an adjusted R^2 should be estimated.

Additionally, some diagnostic tests may be added (as an example, heteroschedasticity and multicollinearity tests), and regression analysis may be replicated on subsets (as an example, by hospital, clinique, office) or on different dependent variables (as an example, the baseline scenario).

Minor comments  

Table 1 may be moved to a dedicated Appendix and may be integrated with a short comment

Author Response

Response to Reviewer 3 Comments

We are grateful to the reviewers for the time and effort he/she spent reviewing our paper. We are also thankful for the insightful comments, which led us to improve our research. Please, find below a response to each comment. For readability purposes, our responses to the comments are in italics. Finally, changes in the manuscript are highlighted in yellow. We hope that this new version meets reviewer´s expectations. The current version of the manuscript has been reviewed by an English native proofreader.

Point 1:  Short Summary

The research illustrates how ACKD does not only affect kidneys, but also patients' biopsychosocial sphere, introducing physical, psychological and socio-labor limitations, and increasing their end-of-life medicalization. To increase the limited knowledge about symptoms and suffering in patients with ACKD undergoing hemodialysis treatment in Ecuador, the research focuses on the most frequent symptoms, not only at the physical but also at the emotional level in the form of suffering, which might be exerting some influence on patients' quality of life. Specifically, the research illustrates the results of a longitudinal study involving a sample of 258 patients (246 respondents), disaggregating by hospital/clinique/service, by a baseline and a two months follow-up scenario, and by a number of concomitant diseases. Finally, a regression model (OLS following a backward stepwise procedure) is presented, explaining the total number of problems at 2-months follow up in terms of socioeconomic and health factors. Results illustrate how suffering in patients with chronic kidney disease is directly related to the symptomatic burden. More in general, chronic kidney disease has been associated to low quality of life, however, it has been proved that palliative care can enhance it.

Response 1:  Thanks for this encouraging comment and your review.

Point 2:  Broad Comments

The research seems to be of interest to the readership of the Journal, however empirical analysis should be improved. Due to its descriptive use, the use of OLS technique is not questionable, but the model specification should be improved. As an example, a correlation analysis among regressors may be added, then the model may be developed on a subset of the available regressors to avoid multicollinearity, at most controlling for (some) socioeconomic factors. Furthermore, the R^2 is low, therefore interaction effects and non linear relations could be introduced to improve the goodness-of-fit. Finally, as there are multiple regressors, an adjusted R^2 should be estimated.

Additionally, some diagnostic tests may be added (as an example, heteroschedasticity and multicollinearity tests), and regression analysis may be replicated on subsets (as an example, by hospital, clinique, office) or on different dependent variables (as an example, the baseline scenario).

Response 2: Thanks for this encouraging comment. We have carried out a correlation analysis, estimated adjusted R^2, and we have replicated the regression analyses on subsets (by hospital, clinic…). Those changes can be reviewed in pages 68.

Point 3:  Minor comments 

Table 1 may be moved to a dedicated Appendix and may be integrated with a short comment

Response 3: Thanks for your comment. We have moved the table 1 to Appendix (pages 1213).

Round 2

Reviewer 3 Report

Broad comments

I would like to thank the Authors for having revised the manuscript according to the comments received, even if the additional work on the empirical model does not fully match my expectations. As an example, I asked to add a correlation matrix because, observing correlations within regressors and among regressors and the dependent variable, provides a selection criterion that leads to drop some variables (those that exhibit a higher correlation with other regressors than with the dependent variable) to have more stable (less volatile) estimates. Furthermore, the adjusted R^2 are of low magnitude, therefore the functional specification of the models may be improved and there may be omitted variables. However, the exercise has mostly a descriptive purpose, and now the Authors provide sufficient information to allow the reader to form her/his own beliefs on the results presented, that anyway are of interest. In brief, improvements of the regression analysis may lead to more robust empirical evidence, but they may be posponed to further researches, as the results presented are reasonable and the estimation technique is well-documented.